# Risk factors for postoperative delirium and subsyndromal delirium in older patients in the surgical ward: A prospective observational study

**Maya Kanno** [1]*, **Mana Doi**[1], **Kazumi Kubota**[2], **Yuka Kanoya**[1]

1 Department of Nursing, Graduate School of Medicine, Yokohama City University, Yokohama, Japan,
2 Department of Healthcare Information Management, The University of Tokyo Hospital, Tokyo, Japan

* mkan@yokohama-cu.ac.jp

**Data Availability Statement:** We did not have participants' permits to share data generated and/or analyzed publicly. Thus, the datasets during the current study are not publicly available. Data are

## Abstract

Postoperative delirium (POD) and subsyndromal delirium (SSD) among older patients is a common, serious condition associated with a high incidence of negative outcomes. However, there are few accurate methods for the early detection of POD and SSD in surgical wards. This study aimed to identify risk factors of POD and SSD in older patients who were scheduled for surgery in a surgical ward. This was a prospective observational study. Study participants were older than 65 years, underwent urology surgery, and were hospitalized in the surgical ward between April and September 2019. Delirium symptoms were assessed using the Confusion Assessment Method (CAM) on the preoperative day, the day of surgery, and postoperative days 1–3 by the surgical ward nurses. SSD was defined as the presence of one or more CAM criteria and the absence of a diagnosis of delirium based on the CAM algorithm. Personal characteristics, clinical data, cognitive function, physical functions, laboratory test results, medication use, type of surgery and anesthesia, and use of physical restraint and bed sensor were collected from medical records. Multiple logistic regression analyses were conducted to identify the risk factors for both POD and SSD. A total of 101 participants (mean age 74.9 years) were enrolled; 19 (18.8%) developed POD (n = 4) and SSD (n = 15). The use of bed sensors (odds ratio 10.2, $p$ = .001) was identified as a risk factor for both POD and SSD. Our findings suggest that the use of bed sensors might be related to the development of both POD and SSD among older patients in surgical wards.

## Introduction

Delirium, an acute fluctuating attention and cognition, is a common and serious condition among older patients. Older age and surgery are risk factors for delirium [1, 2]. The incidence of postoperative delirium (POD) in the older population ranges from 11% to 51% [2] and is associated with many negative outcomes, including the high risk of complications, cognitive

available on request from the Institutional Review Board of the Medical Department of the Yokohama City University (rinri@yokohama-cu.ac.jp) for researchers who meet the criteria for access to data.

**Funding:** The authors received no specific funding for this work.

**Competing interests:** The authors have declared that no competing interests exist.

decline, prolonged hospital stay, rehospitalization, institutionalization, and mortality [2–4]. Therefore, early detection of POD—among older patients—and appropriate care are important to prevent the worsening of POD and the associated complications.

Subsyndromal delirium (SSD) is an important predictor of POD. SSD is characterized by certain symptoms of delirium without the full symptoms, and it is likely to develop into full delirium syndrome [5, 6]. The incidence of postoperative SSD is 68% in orthopedic surgery [7, 8], 30.7% to 37.8% in cardiac surgery [6, 9, 10], and 36.7% in abdominal surgery [11]. Furthermore, SSD, as well as POD, in older patients is also associated with many negative outcomes, including a decline in activities of daily living (ADL), prolonged hospital stay, and high mortality [5, 6, 12, 13]. POD and SSD in older patients have been studied in Europe and America [5–10] and South America [11] but not in Asia, where the population is aging. Therefore, the identification of older patients at risk of postoperative SSD is meaningful in Japan that is becoming a super-aged society in Asia.

For POD, altered cognitive state [14, 15], impaired ADL [14], and anxiety [16] are some of the known risk factors. These factors are important aspects for assessing the ability to perform daily living activities in older adults. In contrast, only three risk factors for SSD have been identified in the surgical ward in previous studies: higher pain level, a recent history of falls within the past 6 months, and a longer preoperative fasting time [7, 8]. We predicted that there are more risk factors for SSD, similar to POD, such as cognitive state, ADL, and anxiety. Thus, the risk factors of SSD in older patients admitted to the surgical ward need to be investigated further. In addition, we need to understand both risk factors for POD and SSD because the concepts of POD and SSD are in continuous development [12]. Therefore, the aim of this study was to identify risk factors for both POD and SSD in older patients who underwent surgery and were admitted postoperatively in the surgical ward.

## Materials and methods

### Study design and participants

This single-center prospective observational study recruited older patients (>65 years) who underwent surgery under either general or spinal anesthesia for urological diseases and were postoperatively treated in the surgical ward of Yokohama City University Medical Center between April 2019 and September 2019. Eligible patients were enrolled if they provided consent for study participation. The exclusion criteria were (1) ICU admission, (2) low level of consciousness before the surgery (Japan Coma Scale score 100–300), (3) inability to speak Japanese, (4) impaired judgment because of developmental disorders or cognitive decline, and (5) preoperative onset of delirium. Participants with missing data for the dependent variables were excluded from the analysis.

### Study procedures

The ward nurses assessed all patients for POD and SSD for 5 days—the day before the surgery, the day of the surgery, and three consecutive days post-surgery—using the Japanese version of the Confusion Assessment Method (CAM), which was developed using the diagnostic criteria specified in the Diagnostic and Statistical Manual of Mental Disorders-III [17] which enabled rapid screening for delirium. We defined the assessment periods in this study based on prior studies [18]. On the day before the surgery, CAM was applied once during the day shift (from hospitalization until 17:00). Postoperatively, patient evaluation using CAM was undertaken three times a day, once during each shift; day shift (9:00 to 17:00), evening shift (17:00 to 1:00), and night shift (1:00 to 9:00). The ward nurses received training that imparted basic knowledge on the identification of POD symptoms and learned how to use CAM to ensure consistency in

assessments by a researcher. The training was designed to minimize the burden on the ward nurses with reference to the Short CAM Training Manual [19], was consistent with a previous study [20], and included (1) the presentation, (2) a SHORT CAM POST-TEST in accordance with the instructions in the Short CAM Training Manual, and (3) an assessment for delirium in three situations, provided by a case presentation video and discussion.

## Measurement of study variables

**Outcomes: Incidence of POD and SSD.** We calculated the incidence of (1) both POD and SSD, (2) separately of POD and SSD to describe their characterization. However, we specified the incidence of both POD and SSD as main outcomes because SSD episodes are closely related to POD [5, 6] and continuous concept to delirium [12]. Patients were evaluated for delirium symptoms using CAM, which comprises four criteria: (1) acute onset and fluctuating course, (2) inattention, (3) disorganized thinking, and (4) altered level of consciousness. The CAM algorithm for the diagnosis of delirium requires the presence of both the first and the second criteria and either the third or the fourth criteria [21]. In this study, POD was defined by the diagnosis of delirium based on the CAM algorithm. The SSD was defined as the presence of one or more CAM criteria and the absence of a diagnosis of delirium based on the CAM algorithm [6, 14, 22, 23]. CAM can be completed in less than 5 minutes [24], and the Japanese version of CAM has a higher sensitivity (83.3%) and specificity (97.6%) when validated for use by nurses compared with psychiatrists [25]. We obtained permission to use the Japanese version of CAM from the copyright holder (Hospital Elder Life Program) and the developer (Akira Watanabe).

**Demographic characteristics and surgical clinical variables.** The demographic characteristics and surgical clinical variables were defined as independent variables, and the Comprehensive Geriatric Assessment-short version (CGA7) was set as the key independent variable. We obtained data on age, sex, ADL-function, comorbidity, history of dementia and cerebrovascular disease, medication use, emergency admission, visual and hearing disabilities, the degree of dementia and care need, the applicable to CGA7 at baseline, and the lesion site. We gathered information as surgical clinical variables regarding the operative method, anesthesia type, operative duration, intraoperative blood loss, preoperative and postoperative results of laboratory blood tests [white blood cell (WBC), red blood cell, hemoglobin (Hb), hematocrit (Ht), platelet (Plt), total protein, albumin (Alb), creatinine (Cr), sodium (Na), potassium (K), chlorine (Cl), calcium (Ca), and C-reactive protein (CRP)], the use of narcotic analgesics, the use of physical restraint, and bed sensors that were recorded on all 5 days. We evaluated physical restraint and bed sensors separately. Physical restraint refers to the use of devices, including belts, ropes, and mittens, to immobilize a person or restrict the ability to move parts of their body freely [26]. In contrast, bed sensors are pressure sensors built into the mattress, which relay on-off signals to the nursing call bell system. Timing of alert by bed sensors can be changed for each activity, such as getting up, sitting on the bed, and getting out of the bed.

ADL was assessed using the Barthel Index, which is an objective scale to evaluate ADL with scores ranging from 0 to 100; a higher score indicates greater independence [27]. Comorbidity was assessed using the Charlson Comorbidity Index (CCI), which is a severity classification scoring tool for comorbidities (0 to 37); a higher score indicates worse illness [28]. Both tools have good reliability and validity.

The degree of dementia and need for care were assessed considering the independence degree of daily living for the older people with dementia and the stage of long-term care need, which are used to determine the appropriate care requirements of older adults, developed by the Ministry of Health, Labour, and Welfare of Japan. Independence degree of daily living for

the older people with dementia to perform ADL has five levels: I, II, III, IV, and M, with M indicating maximum dependence. The stage of long-term care needs has seven levels: support needed (1 and 2) and care needed (1 to 5); care needed 5 indicates maximum dependence.

CGA7 is a screening tool that extracted seven key items from a total of 40 items of four validated scales: the Barthel Index, the revised version of Hasegawa's Dementia Scale, the Vitality Index, and the Geriatric Depression Scale [29]. CGA7 assesses aspects of the geriatric physical, psychological, and social domains. It comprises the following questions: CGA1 (motivation): "Can the subject greet the examiner by himself/herself?", CGA2 (cognitive function): "Can the subject repeat 'cherry blossoms, cats, trains'?," CGA3 (instrumental ADL): "Can the subject go to the hospital by himself/herself?," CGA4 (cognitive function): "Can the subject recall three words in CGA2 and repeat that?," CGA5 (ADL): "Can the subject take a bath by himself/herself?," CGA6 (ADL): "Can the subject use the toilet by himself/herself?," and CGA7 (emotion/mood): "Does the subject feel he/she is powerless?"). CGA7 is assessed with "can"/ "yes" or "cannot"/ "no" for each question; negative outcomes on CGA7 indicate older adults with poor ability to perform daily living activities. The specific assessment of the reliability and validity of the CGA7 test was deemed unnecessary because all four scales from which the tool was compiled have good reliability and validity.

## Statistical analyses

In the primary analysis of this study, we specified the key independent variables from CGA7, which comprised seven items. To calculate the sample size, the lower incidence rates in the number of patients developing POD and SSD and those who did not develop POD and SSD require at least 6 to 10 patients per independent variable for binary logistic regression analysis [30]. A prior study reported POD and SSD incident rates to be 58.1% among older patients undergoing surgery [6]. Thus, we calculated that a minimum of 100 participants were required for the primary analysis.

We calculated two patterns as separate the incidence of POD and SSD and both the incidence of POD and SSD. Continuous variables were analyzed using the Mann–Whitney U test or Kruskal-Wallis test. Categorical variables were analyzed using the chi-square test or Fisher's exact test.

The binary logistic regression analyses (Forward Selection: Likelihood Ratio) of both POD and SSD as the primary analysis was carried out with the key independent variables in each of the seven CGA7 items. The secondary analysis incorporated the total CGA7 (to meet over one of each seven negative CGA7 items or not) and related variables of the incidence of both POD and SSD. Before the binary logistic regression analysis, data were analyzed using univariate analysis to identify the factors related to the incidence of both POD and SSD. Variables with a $p$-value < 0.2 in the univariate analysis were included in the secondary analysis. Additionally, we conducted multinomial logistic regression analysis with separately POD and SSD. All data were analyzed in SPSS statistics version 26, and the significance level was set at $p < 0.05$.

## Ethics approval and consent to participate

This study was approved by the Institutional Review Board of the Medical Department of the Yokohama City University on April 1, 2019 (approval no. B190200026). All participants provided formal written informed consent prior to study participation.

## Results

### Study population

A total of 121 participants were recruited; of these, 20 were excluded and 101 were included in the final analysis (Fig 1). The demographic characteristics of participants are shown in Table 1.

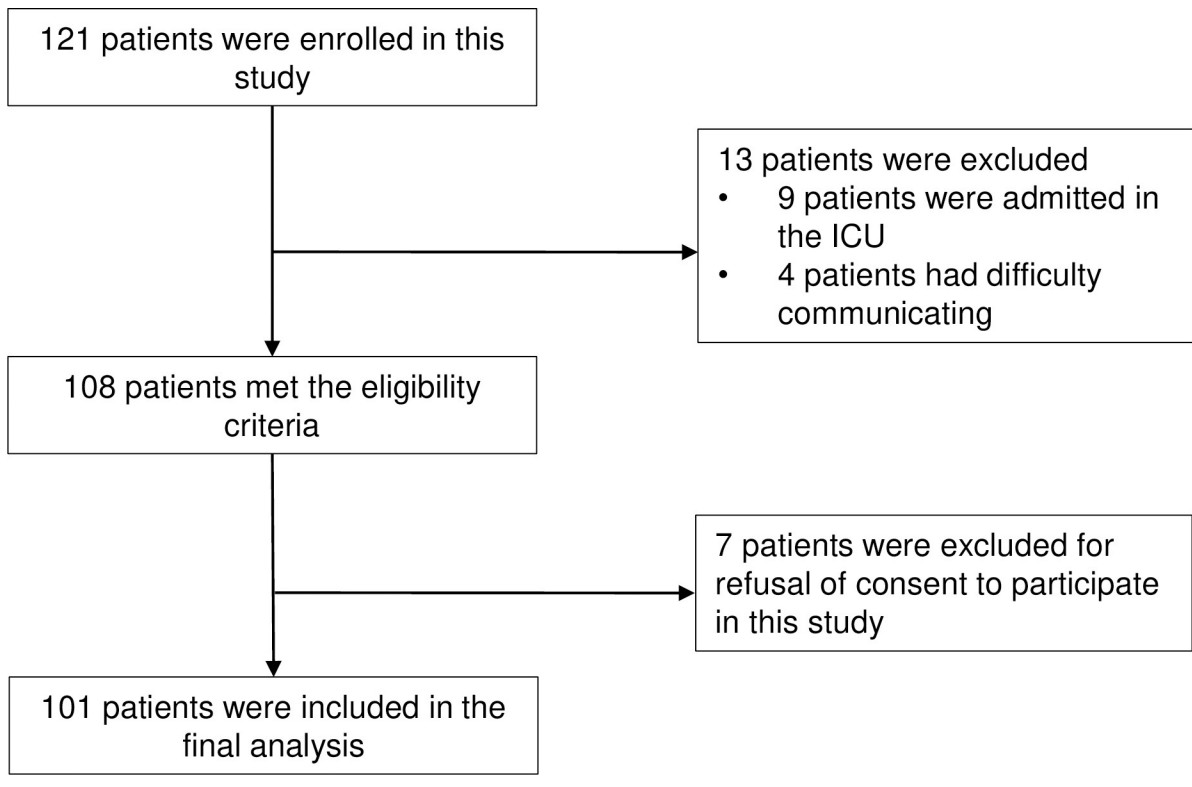

**Fig 1. Patient enrollment flowchart.**

The mean age of patients in this study population was 74.9 (6.2) years, and the majority were male (88.1%). The mean Barthel Index score was 97.7 (9.9) and the mean CCI score was 2.3 (1.1). Most participants had no dementia (98.0%) and were independent. A few participants had a negative total CGA7 (8.9%). Approximately half of the participants had bladder cancer.

### Frequency of POD and SSD

A total of 19 participants (18.8%) were postoperatively observed for POD and SSD using CAM (POD, n = 4; SSD, n = 15). The frequencies of POD and SSD at each evaluation point using CAM are shown in Table 2. The highest frequencies of both POD and SSD were observed on the evening of the day of the surgery. SSD was continuously observed until postoperative day 3.

### Differences in the indicators and POD and SSD (both and separately) in univariate analysis

The incidences of both POD and SSD were significantly associated with high scores of CCI ($p = 0.023$), dementia ($p = 0.034$), emergency admission ($p = 0.034$), large number of medications ($p = 0.016$), and low independence degree of daily living for the older people with dementia ($p = 0.006$) in terms of demographic characteristics (Table 3). Regarding the surgical clinical variables (Table 4), the incidences of both POD and SSD were significantly associated with preoperative high CRP levels ($p = 0.030$), physical restraint use ($p = 0.034$), and bed sensor use ($p = 0.001$).

The separate incidences of POD and SSD were significantly associated with high scores of CCI ($p = 0.041$), dementia ($p = 0.013$), emergency admission ($p = 0.001$), low independence degree of daily living for the older people with dementia ($p = 0.004$), negative outcome in

**Table 1. Patient demographic characteristics (n = 101).**

| Variables | n (%) |
|---|---|
| Age (years), mean (SD) | 74.9 (6.2) |
| Men, n (%) | 89 (88.1) |
| Barthel Index, mean (SD) | 97.7 (9.9) |
| CCI, mean (SD) | 2.3 (1.1) |
| Dementia, n (%) | 2 (2.0) |
| Cerebrovascular disease, n (%) | 6 (5.9) |
| Emergency admission, n (%) | 2 (2.0) |
| Number of medications, mean (SD) | 4.1 (3.5) |
| Anticholinergic drug use, n (%) | 5 (5.0) |
| Diazepam conversion, mean (SD) | 0.8 (2.2) |
| Non-visual disorder, n (%) | 16 (15.8) |
| Non-hearing disorder, n (%) | 71 (70.3) |
| Independence daily living for the older people with dementia, n (%) | 98 (97.0) |
| Independence long-term care need, n (%) | 94 (93.1) |
| Negative outcome in the total CGA7, n (%) | 9 (8.9) |
| CGA1[a] | 1 (1.0) |
| CGA2[b] | 1 (1.0) |
| CGA3[c] | 5 (5.0) |
| CGA4[d] | 1 (1.0) |
| CGA5[e] | 5 (5.0) |
| CGA6[f] | 1 (1.0) |
| CGA7[g] | 3 (3.0) |
| Lesion site, n (%) | |
| Kidney tumor | 10 (9.9) |
| Bladder cancer | 50 (49.5) |
| Prostatic cancer | 16 (15.8) |
| Benign prostatic hypertrophy | 13 (12.9) |
| Others | 12 (11.9) |

Abbreviations: CCI, Charlson Comorbidity Index; CGA7, Comprehensive Geriatric Assessment-short version; SD, standard deviation.

[a]CGA1, Can the subject greet the examiner by himself/herself?

[b]CGA2, Can the subject repeat 'cherry blossoms, cats, trains'?

[c]CGA3, Can the subject go to the hospital by himself/herself?

[d]CGA4, Can the subject recall three words in CGA2 and talk that?

[e]CGA5, Can the subject take a bath by himself/herself?

[f]CGA6, Can the subject use the toilet by himself/herself?

[g]CGA7, Does the subject feel he/she is powerless?

CGA7 ($p = 0.047$), and negative CGA6 ($p = 0.040$) in terms of demographic characteristics (Table 5). Regarding surgical clinical variables (Table 6), the separate incidences of POD and SSD were significantly associated with the postoperative use of sleeping pills ($p = 0.022$), physical restraint use ($p = 0.001$), and bed sensor use ($p<0.001$).

## Risk factors for POD and SSD determined using binary logistic regression analysis

The binary logistic regression analyses of both POD and SSD as the primary analysis were carried out with the key independent variables in each of the seven CGA7 items. However, the

**Table 2. Frequencies of POD and SSD, n (%).**

| | | POD | SSD |
|---|---|---|---|
| The day of the surgery | day | 2 (2.0) | 5 (5.0) |
| | evening | 4 (4.0) | 8 (7.9) |
| Postoperative day 1 | night | 1 (1.0) | 3 (3.0) |
| | day | 0 (.0) | 2 (2.0) |
| | evening | 0 (.0) | 4 (4.0) |
| Postoperative day 2 | night | 1 (1.0) | 1 (1.0) |
| | day | 0 (.0) | 2 (2.0) |
| | evening | 0 (.0) | 1 (1.0) |
| Postoperative day 3 | night | 0 (.0) | 2 (2.0) |
| | day | 0 (.0) | 2 (2.0) |
| | evening | 0 (.0) | 1 (1.0) |
| Total | | 4 (4.0) | 15 (14.9) |

Abbreviations: SSD, subsyndromal delirium POD, postoperative delirium.

model did not work owing to the separation variables [31]. Secondary analysis used the binary logistic regression analysis with the total CGA7 and eight independent variables (i.e., CCI, medication use, preoperative WBC, preoperative Na, preoperative Cl, preoperative CRP, operative duration, and bed sensor use); these showed correlations in the univariate analysis ($p$-value <0.2). As a result of the secondary analysis, it was shown that the incidence of both POD and SSD was associated with bed sensor use [odds ratio (OR) 10.2; $p$ = 0.001, 95% confidence interval (CI) 2.68–38.65] (Table 7). Multinomial logistic regression analyses with separately POD and SSD could not be performed.

## Discussion

This prospective observational study on urological older patients indicates that POD and SSD are related to bed sensor use.

In this study, the incidence of both POD and SSD was 18.8%, compared with 43.4% to 84.9% that was reported previously [6–11]. Moreover, in this study, the incidence of POD was 3.0%, compared with 11%–51% that was reported previously [2]. The incidence of both POD and SSD was remarkably low in comparison to that reported in previous studies. Three reasons may account for this difference. First, it may be due to the remarkably low incidence of patients with dementia. Out of 101 participants, only two patients had a previous diagnosis of dementia. Dementia is a risk factor for POD [15] and SSD [32, 33]. As a result, patients with dementia were excluded from this study. It may have affected the incidence rate of both POD and SSD in this study population. Second, our study participants had a high level of independence. Low ADL is a risk factor for SSD [14, 34]. Thus, a high ADL caused a lower incidence of both POD and SSD. Third, there might have been a selection bias concerning the surgical patients. Since low-performance status and high comorbidity are likely to reduce patient overall survival after surgery, healthcare providers might have preferentially selected healthy older patients for safe surgery [35]. In addition, these patients underwent minimally invasive surgery. Transurethral resection constituted 60.4% of the surgeries, and abdominal surgery was not included in this study. This may have affected the incidence of both POD and SSD.

There was not much difference in the types of relative indicators between both POD and SSD and separately POD and SSD in this study. However, relative indicators were worse for POD, followed by SSD and no delirium. The association of psychiatric symptoms after surgery

**Table 3. Differences in demographic characteristics (no delirium vs. POD and SSD).**

| | No delirium, n = 82 | POD and SSD, n = 19 | *p*-value |
|---|---|---|---|
| Age (years), mean (SD) | 74.8 (6.9) | 75.7 (6.9) | 0.623 |
| Men, n (%) | 51 (62.2) | 16 (84.2) | 0.439 |
| Barthel Index, mean (SD) | 98.1 (9.1) | 94.4 (16.0) | 0.329 |
| CCI, mean (SD) | 2.2 (1.1) | 2.8 (1.4) | 0.023 |
| Dementia, n (%) | 0 (0) | 2 (10.5) | 0.034 |
| Cerebrovascular disease, n (%) | 4 (4.9) | 2 (10.5) | 0.315 |
| Emergency admission, n (%) | 0 (0) | 2 (10.5) | 0.034 |
| Number of medications, mean (SD) | 3.7 (4.5) | 5.8 (30.5) | 0.016 |
| Anticholinergic drug use, n (%) | 3 (3.7) | 2 (10.5) | 0.236 |
| Diazepam conversion, mean (SD) | 0.7 (2.0) | 1.2 (3.2) | 0.520 |
| Non-visual disorder, n (%) | 12 (14.6) | 4 (21.1) | 0.495 |
| Non-hearing disorder, n (%) | 59 (72.0) | 12 (63.2) | 0.655 |
| Independence daily living for the older people with dementia, n (%) | 82 (100.0) | 16 (84.2) | 0.006 |
| Independence long-term care need, n (%) | 76 (92.7) | 18 (94.7) | 0.643 |
| Negative outcome in the total of CGA7, n (%) | 6 (7.3) | 3 (15.8) | 0.36 |
| CGA1[a] | 0 (0) | 0 (0) | 0.034 |
| CGA2[b] | 5 (6.1) | 0 (0) | 1.000 |
| CGA3 [c] | 1 (1.2) | 2 (10.5) | 1.000 |
| CGA4 [d] | 1 (1.2) | 0 (0) | 0.236 |
| CGA5 [e] | 3 (3.7) | 2 (10.5) | 1.000 |
| CGA6 [f] | 1 (1.2) | 1 (5.3) | 0.236 |
| CGA7[g] | 3 (3.7) | 1 (5.3) | 0.188 |
| Lesion site, n (%) | | | 0.469 |
| Kidney tumor | 8 (9.8) | 2 (10.5) | |
| Bladder cancer | 42 (51.2) | 8 (42.1) | |
| Prostatic cancer | 12 (14.6) | 4 (21.1) | |
| Benign prostatic hypertrophy | 9 (11.0) | 4 (21.1) | |
| Others | 11 (13.4) | 1 (5.3) | 0.601 |

Abbreviations: CAM, Confusion Assessment Method; CCI, Charlson Comorbidity Index; CGA7, Comprehensive Geriatric Assessment-short version; POD, postoperative delirium; SD, standard deviation; SSD, subsyndromal delirium.

[a]CGA1, Can the subject greet the examiner by himself/herself?

[b]CGA2, Can the subject repeat 'cherry blossoms, cats, trains'?

[c]CGA3, Can the subject go to the hospital by himself/herself?

[d]CGA4, Can the subject recall three words in CGA2 and talk that?

[e]CGA5, Can the subject take a bath by himself/herself?

[f]CGA6, Can the subject use the toilet by himself/herself?

[g]CGA7, Does the subject feel he/she is powerless?"

Continuous variables are analyzed by Mann–Whitney U test.

Categorical variables are analyzed by Fisher's exact test or the chi-square test.

with demographic and clinical variables in this study may support the concept of a continuous characterization of SSD and delirium that was stated in a previous study [12].

Furthermore, the binary logistic regression analysis suggested that bed sensor use was associated with an increased risk of both POD and SSD. Nurses may force not to stand for the

**Table 4. Differences in surgical clinical variables (no delirium vs. POD and SSD).**

| | No delirium, n = 82 | POD and SSD, n = 19 | *p*-value |
|---|---|---|---|
| Operative methods, n (%) | | | |
| Laparoscopic surgery | 24 (29.3) | 5 (26.3) | |
| Transurethral resection | 50 (61.0) | 11 (57.9) | |
| Others | 8 (9.8) | 3 (15.8) | 0.668 |
| Anesthesia, n (%) | | | |
| General anesthesia | 74 (90.2) | 19 (100.0) | |
| Spinal anesthesia | 8 (9.8) | 0 (0) | 0.346 |
| Results of preoperative blood tests, mean (SD) | | | |
| WBC ($10^3$/μL) | 6.1 (1.6) | 7.1 (2.5) | 0.141 |
| Hb (g/dL) | 13.4 (1.8) | 13.0 (2.1) | 0.546 |
| Cr (mg/dL) | 1.0 (0.6) | 1.2 (0.6) | 0.279 |
| Na (mmol/L) | 141.8 (2.2) | 140.3 (2.8) | 0.056 |
| K (mmol/L) | 4.3 (0.4) | 4.4 (0.4) | 0.426 |
| Cl (mmol/L) | 104.8 (2.5) | 103.6 (2.6) | 0.058 |
| Ca (mmol/L) | 9.4 (0.4) | 9.5 (0.5) | 0.305 |
| CRP (mg/L) | 0.3 (0.5) | 1.8 (4.0) | 0.030 |
| Operative duration (min), mean (SD) | 116.0 (113.7) | 141.1 (113.7) | 0.088 |
| Intraoperative blood loss (mL), mean (SD) | 95.0 (234.0) | 55.3 (146.1) | 0.847 |
| Use of narcotic analgesics, n (%) | 22 (26.8) | 6 (31.6) | 0.777 |
| Postoperative physical status, n (%) | | | |
| Inflammation | 1 (1.2) | 1 (5.3) | 0.342 |
| Anemia | 20 (24.4) | 7 (36.8) | 0.269 |
| Undernutrition | 21 (25.6) | 7 (36.8) | 0.395 |
| Electrolyte abnormality | 11 (13.4) | 0 (0) | 0.119 |
| Decline in renal function | 13 (15.9) | 5 (26.3) | 0.322 |
| Postoperative use of sleeping pills, n (%) | 6 (7.3) | 2 (10.5) | 0.643 |
| Use of physical restraint | 0 (0) | 2 (10.5) | 0.034 |
| Bed sensor use | 5 (6.1) | 7 (36.8) | 0.001 |

Abbreviations: Ca, calcium; Cl, chlorine; Cr, creatinine; CRP, C-reactive protein; Hb, hemoglobin; K, potassium; Na, sodium; POD, postoperative delirium; SD, standard deviation; SSD, subsyndromal delirium; WBC, white blood cell.

Continuous variables are analyzed by Mann–Whitney U test.

Categorical variables are analyzed by Fisher's exact test or the chi-square test.

prevention of falls when older patients with sensor beds are getting out of bed. Depending on how the nurses use sensor beds, older patients would be prevented from participating in physical activity. Physical activity in older people beneficially influences brain function based on exercise promoting neurogenesis and synaptogenesis [36]. As delirium is defined as acute brain dysfunction, low activity may affect POD and SSD. In addition, immobilization is one of the risk factors for delirium in older people [37]. Thus, low activity by using bed sensors might make older patients promote predisposes to both POD and SSD. On the other hand, bed sensors are usually used for patients with dementia and poor ADL to prevent falls [38]. Despite no sign of an altered cognitive state and impaired ADL at baseline, we predicted that nurses may have expected these signs during postoperative assessments and intentionally used bed sensors. In addition, frail older patients were likely to change their rooms near the nurse station. Room transfer is also a risk factor for delirium among older patients [39]. There is a possibility that nurses used bed sensors only at night to prevent falls because of darkness, which suggests that

**Table 5. Differences in demographic characteristics (no delirium vs. SSD vs. POD).**

| | No delirium, n = 82 | SSD, n = 15 | POD, n = 4 | *p*-value |
|---|---|---|---|---|
| Age (years), mean (SD) | 74.8 (6.9) | 74.9 (7.3) | 78.8 (4.4) | 0.330 |
| Men, n (%) | 51 (62.2) | 14 (93.3) | 4 (100) | 0.812 |
| Barthel Index, mean (SD) | 98.1 (9.1) | 96.3 (14.2) | 90.0 (20.0) | 0.242 |
| CCI, mean (SD) | 2.2 (1.1) | 2.7 (1.4) | 3.3 (1.0) | 0.041 |
| Dementia, n (%) | 0 (0) | 1 (6.7) | 1 (25.0) | 0.013 |
| Cerebrovascular disease, n (%) | 4 (4.9) | 2 (10.5) | 0 (0) | 0.401 |
| Emergency admission, n (%) | 0 (0) | 0 (0) | 2 (50.0) | 0.001 |
| Number of medications, mean (SD) | 3.7 (4.5) | 5.4 (3.1) | 7.3 (5.1) | 0.050 |
| Anticholinergic drug use, n (%) | 3 (3.7) | 1 (6.7) | 1 (25.0) | 0.119 |
| Diazepam conversion, mean (SD) | 0.7 (2.0) | 1.5 (3.5) | 0 (0) | 0.365 |
| Non-visual disorder, n (%) | 12 (14.6) | 2 (13.3) | 2 (50.0) | 0.223 |
| Non-hearing disorder, n (%) | 59 (72.0) | 9 (60.0) | 3 (75.0) | 0.729 |
| Independence daily living for the older people with dementia, n (%) | 82 (100.0) | 13 (86.7) | 3 (75.0) | 0.004 |
| Independence long-term care need, n (%) | 76 (92.7) | 15 (100.0) | 3 (75.0) | 0.426 |
| Negative outcome in the total of CGA7, n (%) | 6 (7.3) | 1 (6.7) | 2 (50.0) | 0.047 |
| CGA1[a] | 0 (0) | 0 (0) | 0 (0) | 1.000 |
| CGA2[b] | 5 (6.1) | 0 (0) | 0 (0) | 1.000 |
| CGA3[c] | 1 (1.2) | 1 (6.7) | 1 (25.0) | 0.119 |
| CGA4[d] | 1 (1.2) | 0 (0) | 0 (0) | 1.000 |
| CGA5[e] | 3 (3.7) | 1 (6.7) | 1 (25.0) | 0.119 |
| CGA6[f] | 1 (1.2) | 0 (0) | 1 (25.0) | 0.040 |
| CGA7[g] | 3 (3.7) | 0 (0) | 1 (25.0) | 0.170 |
| Lesion site, n (%) | | | | |
| Kidney tumor | 8 (9.8) | 1 (6.7) | 1 (25.0) | |
| Bladder cancer | 42 (51.2) | 6 (40.0) | 2 (50.0) | |
| Prostatic cancer | 12 (14.6) | 4 (26.7) | 0 (0) | |
| Benign prostatic hypertrophy | 9 (11.0) | 4 (26.7) | 1 (25.0) | |
| Others | 11 (13.4) | 0 (0) | 1 (25.0) | 0.274 |

Abbreviations: CAM, Confusion Assessment Method; CCI, Charlson Comorbidity Index; CGA7, Comprehensive Geriatric Assessment-short version; POD, postoperative delirium; SD, standard deviation; SSD, subsyndromal delirium.

[a]CGA1, Can the subject greet the examiner by himself/herself?

[b]CGA2, Can the subject repeat 'cherry blossoms, cats, trains'?

[c]CGA3, Can the subject go to the hospital by himself/herself?

[d]CGA4, Can the subject recall three words in CGA2 and talk that?

[e]CGA5, Can the subject take a bath by himself/herself?

[f]CGA6, Can the subject use the toilet by himself/herself?

[g]CGA7, Does the subject feel he/she is powerless?

Continuous variables are analyzed using the Kruskal-Wallis test.

Categorical variables are analyzed by Fisher's exact test or the chi-square test.

night-time investigations may be associated with an increased delirium incidence [40]. These elements might be potential confounders of the association between the use of bed sensors and the development of both POD and SSD.

## Limitations

This study has several limitations. First, there was a low incidence of POD and SSD in contrast to our expectations; this was because most participants were healthy, and we consequently

**Table 6. Differences in surgical clinical variables (no delirium vs. SSD vs. POD).**

| | No delirium, n = 82 | SSD, n = 15 | POD, n = 4 | *p*-value |
|---|---|---|---|---|
| Operative methods, n (%) | | | | |
| Laparoscopic surgery | 24 (29.3) | 4 (26.7) | 1 (25.0) | |
| Transurethral resection | 50 (61.0) | 9 (60.0) | 2 (50.0) | |
| Others | 8 (9.8) | 2 (13.3) | 1 (25.0) | 0.794 |
| Anesthesia, n (%) | | | | |
| General anesthesia | 74 (90.2) | 15 (100.0) | 4 (100.0) | |
| Spinal anesthesia | 8 (9.8) | 0 (0) | 0 (0) | 0.536 |
| Results of preoperative blood tests, mean (SD) | | | | |
| WBC ($10^3$/μL) | 6.1 (1.6) | 7.1 (2.4) | 7.1 (3.4) | 0.282 |
| Hb (g/dL) | 13.4 (1.8) | 13.4 (1.8) | 11.6 (2.7) | 0.317 |
| Cr (mg/dL) | 1.0 (0.6) | 1.1 (0.6) | 1.3 (0.7) | 0.540 |
| Na (mmol/L) | 141.8 (2.2) | 140.8 (2.4) | 138.5 (3.7) | 0.054 |
| K (mmol/L) | 4.3 (0.4) | 4.3 (0.3) | 4.6 (0.6) | 0.526 |
| Cl (mmol/L) | 104.8 (2.5) | 104.1 (2.5) | 101.5 (2.4) | 0.057 |
| Ca (mmol/L) | 9.4 (0.4) | 9.5 (0.6) | 9.5 (0.3) | 0.541 |
| CRP (mg/L) | 0.3 (0.5) | 1.5 (3.6) | 3.0 ($<$0.1) | 0.095 |
| Operative duration (min), mean (SD) | 116.0 (113.7) | 134.8 (114.2) | 164.8 (81.7) | 0.174 |
| Intraoperative blood loss (mL), mean (SD) | 95.0 (234.0) | 63.3 (163.1) | 25.0 (50.0 | 0.978 |
| Use of narcotic analgesics, n (%) | 22 (26.8) | 4 (26.7) | 2 (50.0) | 0.657 |
| Postoperative physical status, n (%) | | | | |
| Inflammation | 1 (1.2) | 1 (6.7) | 0 (0) | 0.342 |
| Anemia | 20 (24.4) | 5 (33.3) | 2 (50.0) | 0.323 |
| Undernutrition | 21 (25.6) | 5 (33.3) | 2 (50.0) | 0.384 |
| Electrolyte abnormality | 11 (13.4) | 0 (0) | 0 (0) | 0.298 |
| Decline in renal function | 13 (15.9) | 3 (20.0) | 2 (50.0) | 0.191 |
| Postoperative use of sleeping pills, n (%) | 6 (7.3) | 0 (0) | 2 (50.0) | 0.022 |
| Use of physical restraint | 0 (0) | 0 (0) | 2 (50.0) | 0.001 |
| Bed sensor use | 5 (6.1) | 5 (33.3) | 3 (75.0) | $>$.001 |

Abbreviations: Ca, calcium; Cl, chlorine; Cr, creatinine; CRP, C-reactive protein; Hb, hemoglobin, K, potassium; Na, sodium; POD, postoperative delirium; SD, standard deviation; SSD, subsyndromal delirium; WBC, white blood cell.

Continuous variables are analyzed using the Kruskal-Wallis test.

Categorical variables are analyzed by Fisher's exact test or the chi-square test.

excluded patients with dementia who had a high risk of delirium. In effect, we missed potential risk factors. Second, the influence of nursing care could not be evaluated. Good nursing care might affect the incidence of both POD and SSD and other risk factors and suppress the relationship between such factors and the study outcomes. Third, we did not assess cognitive

**Table 7. Risk factors for both POD and SSD in the binary logistic regression analysis.**

| Variables | β | SE | *p*-value | OR | 95% CI |
|---|---|---|---|---|---|
| Bed sensor use No, 0; Yes, 1 | 2.321 | 0.681 | 0.001 | 10.18 | 2.682–38.652 |
| Preoperative CRP mg/dL | 0.388 | 0.202 | 0.054 | 1.47 | 0.993–2.188 |

Abbreviations: 95% CI, 95% confidence interval; CRP, C-reactive protein; POD, postoperative delirium; SE, standard error; SSD, subsyndromal delirium; β, regression coefficient.

functions during CAM assessments concerning the burden of older patients because the cognitive scale required patient independent work. CAM should be scored based on a formal cognitive evaluation to distinguish between delirium and dementia because of their similarity. Thus, there was a possibility that participants with dementia were identified as having delirium. This might have affected the incidence rate of both POD and SSD. In addition, we did not assess the severity of each POD and SSD. The severity of each symptom of CAM's four criteria did not clarify. However, despite these limitations, this study provides a knowledge base for the evaluation of both POD and SSD in older patients in the surgical ward for future studies. Future studies are needed to assess and adjust for relevant confounders for "bed sensor use" and thus confirm relevant risk factors for both POD and SSD.

## Conclusion

The aim of this study was to identify risk factors for both POD and SSD in older patients who underwent surgery and were admitted postoperatively in the surgical ward. A total of 19 participants (18.8%) were postoperatively observed for POD and SSD using CAM (POD, n = 4; SSD, n = 15). The use of bed sensors (odds ratio 10.2, p = .001) was identified as a risk factor for both POD and SSD.

## Acknowledgments

We thank all the participants and the ward nurses who supported this study.

## Author Contributions

**Data curation:** Maya Kanno.

**Formal analysis:** Maya Kanno, Kazumi Kubota.

**Investigation:** Maya Kanno.

**Methodology:** Maya Kanno.

**Supervision:** Mana Doi, Kazumi Kubota, Yuka Kanoya.

**Writing – original draft:** Maya Kanno.

**Writing – review & editing:** Maya Kanno, Mana Doi, Yuka Kanoya.

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
