## [Decision Letter · Decision Letter 0]

9 Apr 2021

PONE-D-21-02796

Risk factors for postoperative delirium and subsyndromal delirium in older patients in the surgical ward: A prospective observational study

PLOS ONE

Dear Dr. Maya,

Thank you for submitting your manuscript to PLOS ONE. After careful consideration, we feel that it has merit but does not fully meet PLOS ONE’s publication criteria as it currently stands. Therefore, we invite you to submit a revised version of the manuscript that addresses the points raised during the review process.

We look forward to receiving your revised manuscript.

Kind regards,

Itamar Ashkenazi

Academic Editor

PLOS ONE

Journal Requirements:

Reviewers' comments:

Reviewer's Responses to Questions

**Comments to the Author**

1. Is the manuscript technically sound, and do the data support the conclusions?

Reviewer #1: Partly

Reviewer #2: Yes

2. Has the statistical analysis been performed appropriately and rigorously? 

Reviewer #1: Yes

Reviewer #2: No

3. Have the authors made all data underlying the findings in their manuscript fully available?

Reviewer #1: Yes

Reviewer #2: Yes

4. Is the manuscript presented in an intelligible fashion and written in standard English?

Reviewer #1: Yes

Reviewer #2: Yes

5. Review Comments to the Author

Reviewer #1: 1.- One of the main limits of the study is NOT to have measured the severity of delirium, and to consider subsyndromal delirium in the same category when there is still little evidence of this entity and no more than the reference is described in the citations (6 ) related to this, it should be clarified whether the same risk factors and the same outcome are expected in subsyndromal delirium.

2.- Despite writing a detailed methodological and systematic process, the calculation of the sample on the incidence of delirium does not make it clear if the results are due to lack of statistical power.

3.- The reason why the use of a bed sensor represents a risk for delirium should be expanded in the discussion in addition to what the authors of the physical restrictions comment, they should explain the biological plausibility of this risk association as it could be secondary immobility which predisposes to delirium.

Reviewer #2: Introduction

1. Introduction is not comprehensive. Needs to reflect what the magnitude of POD and SSD looks globally and specifically in the study area/country?

Materials and methods

2. Study setting was not clearly described.

3. Line 79-83, regarding your exclusion criteria. How did you see the relation between exclusion number 2 and 5. If you excluded preoperative onset of delirium, I think that was enough and no need to exclude specifically low level of consciousness before the surgery, as this one is one of the CAM (preoperative delirium).

4. Way of sample size calculation was not clearly stated. What was your power?

5. Analysis: in the method the authors need to elaborate how the analysis was done like to POD and SSD separately or to both the incidence of POD and SSD merged together?

Results

6. Table 1 and 2 were not cited in the text and also, their given titles were not self-explanatory and needs to re-write in meaningful statements.

7. In table 1 and 2, the authors write “POD and SSD” (n=19). This was not appropriately presented and the authors need to describe the magnitude of POD and SSD separately. If the patient fulfill POD, I think no need to talk about SSD because of SSD is included in POD. You can talk about SSD, if the patient does not fulfill POD as assessed by CAM. Therefore, what does it mean when you say “POD and SSD”? how the patient can be diagnosed with both at once? Or if you have another concern/justification you can justify. For stance, this the delirium is fluctuating condition the patient can be diagnosed with POD at a time and after a certain period of time can be diagnosed with SSD. So how did you manage this condition? Needs more clarification.

8. Generally, the analysis part is confusing and not clearly presented. The proposed analysis methods in method were not clearly documented in tables. E.g., the t-test result was not included in tables. Which logistic regression did you use (linear/ multiple linear logistic regression)?

Discussion

9. Page 251-252, the comparation needs to be with similar articles. In this case you compared with ref no. 6 and 12 which were conducted only on SSD but in your study, there is incidence of both POD and SSD.

6. PLOS authors have the option to publish the peer review history of their article (what does this mean?). If published, this will include your full peer review and any attached files.

Reviewer #1: No

Reviewer #2: **Yes: **Tilahun Abdeta

---

## [Author Response · Author response to Decision Letter 0]

17 Jun 2021

Response to Decision Letter

RESPONSE TO REVIEWER 1

We appreciate the time and effort Reviewer 1 has dedicated to providing insightful comments on ways to improve our paper. We agree with all the suggestions made by Reviewer 1 and have revised the manuscript accordingly. To recognize the differences made, we have highlighted the changes in yellow.

1. One of the main limits of the study is NOT to have measured the severity of delirium, and to consider subsyndromal delirium in the same category when there is still little evidence of this entity and no more than the reference is described in the citations (6) related to this, it should be clarified whether the same risk factors and the same outcome are expected in subsyndromal delirium.

Response: Thank you for your comments. Indeed, we did not measure severity but incidence of delirium, and we placed SSD and POD into the same category. As you have pointed out, we did not assess severity of each POD and SSD, which we now added to the limitation section. In fact, three of four participants with POD showed symptoms of SSD. SSD was observed both before and after developing POD. We described these as the same category because we considered that it is hard to separate POD from SSD into two different and independent phenomena. However, we agree with your suggestion that it needs to be verified whether the same results can be obtained by assessing SSD alone; we now showed SSD result alone and a combined POD and SSD result. This is now added to the Results section in the revised manuscript.

2. Despite writing a detailed methodological and systematic process, the calculation of the sample on the incidence of delirium does not make it clear if the results are due to lack of statistical power.

Response: We are grateful for your pointing this out and making these suggestions. We aimed for 6-10 events per variable to motivate the sample size in this study. Follow suggestion, we calculated a post hoc power. A sample of 101 participants was estimated as being capable of providing sufficient statistical power of 0.96 with α = .05. 

3. The reason why the use of a bed sensor represents a risk for delirium should be expanded in the discussion in addition to what the authors of the physical restrictions comment, they should explain the biological plausibility of this risk association as it could be secondary immobility which predisposes to delirium. 

Response: We totally agree with your valuable advice. Your suggestion had given our discussion a lot of academic insight. We have revised the text as follow to the Discussion section. Page 30, lines 330-335.

” Physical activity in older people beneficially influences brain function based on exercise promoting neurogenesis and synaptogenesis. As delirium is defined as acute brain dysfunction, low activity may affect POD and SSD. In addition, immobilization is one of the risk factors for delirium in older people. Thus, low activity by using bed sensors might make older patients promote predisposes to both POD and SSD.”

RESPONSE TO REVIEWER 2

We wish to express our sincere appreciation to Reviewer 2 for critical reading our manuscript. We totally agree with the suggestions made by Reviewer 2 and have revised our manuscript according to these suggestions. To recognize the differences between the newly added text easily, we have highlighted the changes in yellow.

Introduction

1. Introduction is not comprehensive. Needs to reflect what the magnitude of POD and SSD looks globally and specifically in the study area/country?

Response: You have raised important insights. As you suggested, we added introduction the meaningful to study POD and SSD among older patients in Japan and the need of research widely. Page 4, lines 58-62.

Materials and methods

2. Study setting was not clearly described.

Response: Thank you for your comment. We have added this information. Page 5, lines 80-81.

3. Line 79-83, regarding your exclusion criteria. How did you see the relation between exclusion number 2 and 5. If you excluded preoperative onset of delirium, I think that was enough and no need to exclude specifically low level of consciousness before the surgery, as this one is one of the CAM (preoperative delirium).

Response: We appreciate your comment. As you have pointed out, low level of consciousness is one of the criteria of the CAM. However, "4 altered level of consciousness", that is exclusion criteria (2) low level of consciousness before the surgery, must not need for the diagnosis of delirium. Therefore, we set both exclusion criteria (2) low level of consciousness before the surgery and (5) preoperative onset of delirium.

4. Way of sample size calculation was not clearly stated. What was your power?

Response: We are grateful for your pointing this out and making these suggestions. We aimed for 6-10 events per variable to motivate the sample size in this study. Follow suggestion, we calculated a post hoc power. A sample of 101 participants was estimated as being capable of providing sufficient statistical power of 0.96 with α = .05. 

5. Analysis: in the method the authors need to elaborate how the analysis was done like to POD and SSD separately or to both the incidence of POD and SSD merged together?

7. Results: In table 1 and 2, the authors write “POD and SSD” (n=19). This was not appropriately presented and the authors need to describe the magnitude of POD and SSD separately. If the patient fulfill POD, I think no need to talk about SSD because of SSD is included in POD. You can talk about SSD, if the patient does not fulfill POD as assessed by CAM. Therefore, what does it mean when you say “POD and SSD”? how the patient can be diagnosed with both at once? Or if you have another concern/justification you can justify. For stance, this the delirium is fluctuating condition the patient can be diagnosed with POD at a time and after a certain period of time can be diagnosed with SSD. So how did you manage this condition? Needs more clarification.

Response: Thank you for your advice. We had not separated SSD from POD in the previous version of the manuscript. In fact, three of four participants with POD showed symptoms of SSD. SSD was observed both before and after developing POD. We described POD and SSD as belonging to the same category because we considered that it was hard to distinguish between POD and SSD as completely different phenomena. However, we agree with the Reviewer 2 that it is important to describe the magnitudes of each POD and SSD separately. We have now added separate incidences of each POD and SSD as well as the combined incidence of both POD and SSD to the Results section. In addition, the binary logistic regression analyses of both POD and SSD as the primary analysis was carried out with the key independent variables in each seven CGA7 items. However, the model did not work owing to the separation variables. Also, multinomial logistic regression analysis with separately POD and SSD did not work. Thus, we conducted a secondary analysis using the binary logistic regression analysis with total CGA7 and eight independent variables (e.g., CCI, medication use, preoperative WBC, preoperative Na, preoperative Cl, preoperative CRP, operative duration, and bed sensor use); these showed correlations in the univariate analysis (p-value <0.2). We revised the Statistical analyses and Result section.”

Results

6. Table 1 and 2 were not cited in the text and also, their given titles were not self-explanatory and needs to re-write in meaningful statements.

Response: We appreciate your comment and suggestions. We had revised table titles to improve the clarity and cited them in article according to your suggestion.

8. Generally, the analysis part is confusing and not clearly presented. The proposed analysis methods in method were not clearly documented in tables. E.g., the t-test result was not included in tables. Which logistic regression did you use (linear/ multiple linear logistic regression)?

Response: Thank you for your comment and questions. We used binary logistic regression and revised Methods section to add further clarifications on the statistical analysis, result section, and table with the references to your previous comments 5 and 7.

Discussion

9. Page 251-252, the comparation needs to be with similar articles. In this case you compared with ref no. 6 and 12 which were conducted only on SSD but in your study, there is incidence of both POD and SSD.

Response: We agree with your assessment. However, no. 6 and 12 also studied POD in each of them. We have added another reference that investigated both POD and SSD to include wider range of previous findings for comparison.

---

## [Decision Letter · Decision Letter 1]

21 Jul 2021

Risk factors for postoperative delirium and subsyndromal delirium in older patients in the surgical ward: A prospective observational study

PONE-D-21-02796R1

Dear Dr. Maya,

We’re pleased to inform you that your manuscript has been judged scientifically suitable for publication and will be formally accepted for publication once it meets all outstanding technical requirements.

Kind regards,

Itamar Ashkenazi

Academic Editor

PLOS ONE

Reviewers' comments:

Reviewer's Responses to Questions

**Comments to the Author**

1. If the authors have adequately addressed your comments raised in a previous round of review and you feel that this manuscript is now acceptable for publication, you may indicate that here to bypass the “Comments to the Author” section, enter your conflict of interest statement in the “Confidential to Editor” section, and submit your "Accept" recommendation.

Reviewer #1: All comments have been addressed

2. Is the manuscript technically sound, and do the data support the conclusions?

Reviewer #1: Yes

3. Has the statistical analysis been performed appropriately and rigorously? 

Reviewer #1: Yes

4. Have the authors made all data underlying the findings in their manuscript fully available?

Reviewer #1: Yes

5. Is the manuscript presented in an intelligible fashion and written in standard English?

Reviewer #1: Yes

6. Review Comments to the Author

Reviewer #1: The authors have responded adequately to the recommendations and the work has been substantially improved

7. PLOS authors have the option to publish the peer review history of their article (what does this mean?). If published, this will include your full peer review and any attached files.

Reviewer #1: **Yes: **Sara G. Aguilar Navarro

---

## [Editor Report · Acceptance letter]

23 Jul 2021

PONE-D-21-02796R1 

Risk factors for postoperative delirium and subsyndromal delirium in older patients in the surgical ward: A prospective observational study 

Dear Dr. Kanno:

I'm pleased to inform you that your manuscript has been deemed suitable for publication in PLOS ONE. Congratulations! Your manuscript is now with our production department. 

Kind regards, 

on behalf of

Dr. Itamar Ashkenazi 

Academic Editor

PLOS ONE